# Where talent flows: Trends and determinants of Chinese students' city preferences

**Li Wang[1]☉, Xian Zhang[2]☉, Yifei Wang[3], Yuxiang Li [iD][3]***

**1** School of Social and Behavioral Sciences, Nanjing University, Nanjing, China, **2** University of International Business and Economics, Shenzhen Institute, Shenzhen, China, **3** School of Life Sciences, Nanjing University, Nanjing, China

☉ These authors contributed equally to this paper.
* YuxiangLi@nju.edu.cn

## Abstract

Employment location preferences offer critical insights into how students evaluate opportunities across cities and drive early-career spatial sorting. This study examines Chinese college students' employment city preferences from 2016 to 2020, focusing on the evolving influence of campus performance, family background, and university characteristics. Using five waves of nationally representative longitudinal data from the Panel Study of Chinese University Students (PSCUS), we apply multinomial logistic regression and relative importance analyses. Our results show that university characteristics as the primary predictor: students from Double First-Class universities favor first-tier cities, even as second-tier cities attract the largest share of students. Strong campus performance allows students to be more flexible in their location choices, whereas family background plays a moderate, compensatory role. Temporal patterns suggest that students' preferences reflect the combined influence of institutional prestige, personal merit, and family resources. The findings provide insights into early-career mobility and inform policies for balanced regional development.

## Introduction

Global talent competition has become a focal point in migration research, shifting attention to how highly educated individuals weigh risks and opportunities during their career location decisions [1–3]. College students represent a crucial demographic whose early-career preferences influence both individual life trajectories and the spatial allocation of human capital [4,5]. These choices therefore constitute key behavioral decisions with implications for regional development [6–8].

In rapidly urbanizing countries like China, students' employment location preferences matter greatly [9]. These movements drive important talent flows, defined as the spatial redistribution of highly educated individuals across cities, which influence

**Data availability statement:** The data used in this study are from the Chinese University Student Survey (PSCUS), which is administered by the Institute of Sociology, Chinese Academy of Social Sciences. The raw data are third-party data and are not publicly available for redistribution by the authors.

**Funding:** The author(s) received no specific funding for this work.

**Competing interests:** The authors have declared that no competing interests exist.

urban competitiveness, labor market vitality, and long-term economic resilience [10,11]. Furthermore, these movements carry broad social implications for equity and sustainability, as concentrated flows of human capital may intensify regional disparities [12].

Recent studies highlight several trends in talent flows in China, including the enduring pull of first-tier cities, the growing attractiveness of second-tier cities, and the expanding role of local talent-attraction policies [13–16]. Since 2018, second-tier cities such as Wuhan, Xi'an, and Nanjing have joined the "talent wars," implementing incentives such as housing subsidies, household registration (*Hukou*) reforms, and employment support [17]. These interventions aim to reshape students' opportunity perceptions and reduce uncertainties associated with relocating beyond top-tier cities [18]. Yet the actual effectiveness of these policies in shifting students' preferences remains debated [19].

To move beyond simply tracking where students go, migration and human capital studies emphasize how individual cognition, social resources, and institutional contexts shape these preferences [20]. Three conceptually distinct mechanisms help explain this process. First, the spatial sorting hypothesis (grounded in human capital theory) posits that high-achieving students are disproportionately attracted to first-tier cities to maximize returns on their skills [21–23], leveraging institutional prestige as a strong labor-market signal [24]. Second, the compensatory resource hypothesis (derived from social capital theory) suggests that family background buffers the high risks and costs of migrating to competitive urban centers, especially for students lacking elite credentials [25–27]. Finally, the policy-induced preference hypothesis proposes that local interventions such as *Hukou* reforms and housing subsidies, reshape perceived opportunity structures and make second-tier cities more attractive alternatives for non-elite students [17,28]. Together, these perspectives provide a multidimensional framework for understanding how students navigate a stratified urban hierarchy.

Existing studies have linked city preferences to academic achievement, family background, university prestige, and urban characteristics [29–32]. However, many rely on cross-sectional data, limited geographic scopes, or inconsistent city-tiers classifications [17–20]. As a result, the temporal dynamics of students' location preferences remain insufficiently understood, particularly in periods of rapid labor-market transformation and policy experimentation.

Drawing on five waves (2016–2020) of nationally representative data from the Panel Study of Chinese University Students (PSCUS), this study addresses two core questions: (1) How have Chinese college students' employment city preferences changed over time? (2) How has the relative influence of campus performance, family background, and university characteristics changed in shaping these preferences?

Using multinomial logistic regression and relative importance analyses, we provide a systematic assessment of how spatial sorting, compensatory resources, and policy incentives shape students' city preferences over time. The findings contribute to migration literature by clarifying the shifting balance between individual ability, social capital, and institutional contexts in early-career mobility.

## Materials and methods

### Data source

We draw on five waves of data (2016–2020) from the PSCUS (http://www.pscus.cn/), a large-scale national survey conducted by the Institute of Sociology at the Chinese Academy of Social Sciences. The PSCUS adopts a stratified sampling strategy based on university type (e.g., Double First-Class universities, regular undergraduate colleges, and higher vocational institutions), academic orientation (science and engineering, comprehensive, and humanities and social sciences), and geographic region. Because the available data are fully anonymized and lack individual identifiers across waves. We treat the data as pooled person-wave observations, focusing on population-level associations rather than individual-level panel dynamics. Since we used secondary, de-identified data from an institutional survey, human subjects approval and direct participant contact were not required. After excluding cases with missing values, the final analytical sample contains 50267 person-wave observations. A comparison between the raw dataset and the complete-case sample reveals minimal differences in variable distributions (less than 4%, S1 Table), indicating no strong systematic missingness.

### Variables

**Dependent variable.** Previous studies have primarily focused on whether college students prefer to work in first-tier cities [33,34], often overlooking the distinctions between second-tier and smaller cities. To address this gap, we categorize employment cities into three types, representing different opportunity structures and perceived life chances (S2 Table):

• **First-tier cities:** Beijing, Shanghai, Guangzhou, and Shenzhen.

• **Second-tier cities:** Provincial capitals or economically developed non-capital cities.

• **Smaller cities:** Third- and fourth-tier cities and county-level cities.

This classification derives from the PSCUS survey question, "Where would you most like to work after graduation?" Although the response options vary slightly across survey waves, we systematically recoded all responses into a consistent three-tier structure.

**Independent variables.** We examine three domains that may influence students' employment city preferences as migration-related decisions shaped by individual ability, social resources, and institutional contexts:

• **Campus Performance:**

**Academic performance:** A five-level categorical variable ranging from "very poor" to "excellent", reflecting the student's perceived academic competence.

**Leadership experience:** A binary variable indicating whether the student holds a leadership position, reflecting organizational skills and social recognition.

**Extracurricular participation:** A binary variable capturing participation in student organizations, serving as a proxy for social integration and peer influence.

**Party membership:** A binary variable denoting membership in the Chinese Communist Party, signaling institutional trust and political capital.

• **Family Background:**

**Household registration (*Hukou*) type:** Urban vs. rural (binary variable), capturing early-life opportunities and structural constraints.

**Father's education level:** Years of schooling completed by the student's father, reflecting intergenerational human capital.

**Father's employment sector:** A binary indicator of whether the father works in a public institution, serving as a proxy for parental job stability.

**Household income:** Log-transformed annual household income, indicating economic resources and risk tolerance.

**Only-child status:** A binary variable noting whether the student is an only child, representing concentrated parental investment and resource advantages.

• **University Characteristics:**

**University location:** Categorized as a first-tier, second-tier, or smaller city, reflecting localized opportunity structures.

**University type:** Classified into Project "985" institutions, Project "211" institutions, regular undergraduate colleges, and higher vocational institutions, shaping perceived prestige and career pathways.

Our models also control for gender, age, education level (junior college, bachelor, master, or doctor), year of study, and geographic origin. Table 1 presents the descriptive statistics, including the distribution of employment city preferences and key variables. Approximately 39.39% of students prefer first-tier cities, 50.15% choose second-tier cities, and 10.46% prefer smaller cities. Among the institutions, 21.38% of students are enrolled in universities located in first-tier cities, 55.33% in second-tier cities, and 23.30% in smaller cities. Demographically, 46.81% of the sample are male, with an average age of 20.03 years. Notably, 62.65% are only children.

**Table 1. Summary of descriptive statistics (N = 50267).**

| Employment city preferences | | University location | |
|---|---|---|---|
| First-tier cities | 39.39% | First-tier cities | 21.38% |
| Second-tier cities | 50.15% | Second-tier cities | 55.33% |
| Smaller cities | 10.46% | Smaller cities | 23.30% |
| **Academic performance** | | **University type** | |
| Very poor | 2.78% | Project "985" institutions | 24.99% |
| Poor | 6.93% | Project "211" institutions | 23.94% |
| Average | 49.61% | Regular undergraduate colleges | 26.25% |
| Good | 25.12% | Higher vocational institutions | 24.82% |
| Excellent | 15.57% | **Male** | 46.81% |
| **Leadership experience** | 78.10% | **Age** | 20.03 |
| **Extracurricular participation** | 82.10% | **Only-child status** | 62.65% |
| **Degree level** | | **Geographic origin** | |
| Junior college | 33.65% | West | 28.61% |
| Bachelor | 53.65% | East | 36.47% |
| Master | 12.45% | Central | 27.24% |
| Doctor | 0.25% | Northeast | 7.68% |
| **Party membership** | 11.90% | **Urban *Hukou*** | 46.70% |
| **Father's education level** | | **Father in public institutions** | 26.73% |
| Primary | 2.69% | **Year** | |
| Junior high school | 33.64% | 2016 | 17.98% |
| High school | 23.57% | 2017 | 23.65% |
| Junior college | 25.90% | 2018 | 24.82% |
| Bachelor | 11.73% | 2019 | 22.97% |
| Master+ | 2.47% | 2020 | 10.58% |

**Note**: Categorical variables are presented as proportions for each category, and continuous variables are presented as means values.

## Statistical analysis

We employ multinomial logistic regression models [35] to examine the factors associated with students' employment city preferences. This modeling framework allows for the simultaneous comparison of multiple predictors across city categories. To assess potential multicollinearity among independent variables, we calculate the generalized variance inflation factor (GVIF). Variables with high collinearity are excluded sequentially: university location is removed due to strong collinearity with university type, and age is excluded while retaining education level, resulting in GVIF-adjusted values below 2 for all remained predictors (S3 Table). In year-specific regressions, collinearity between education level and university type is occasionally high, in which case education level is removed (S4 Table). We evaluate the Independence of Irrelevant Alternatives (IIA) assumption using the Hausman–McFadden test [36]. Test statistics are non-significant for all alternative-specific comparisons, indicating that removing any single employment-city category does not systematically alter the estimated coefficients. This result supports the use of the multinomial logistic regression model in this context. To ensure temporal robustness, we ran separate regressions for each survey wave. The direction and relative magnitude of key predictors remain largely stable across years (S5-S9 Tables).

Finally, to compare effect sizes, we calculate standardized coefficients for all predictors. To evaluate the relative importance of the three domains (campus performance, family background, and university characteristics), we average the absolute standardized coefficients within each group. These aggregated measures allow for a clearer comparison of each domain's overall contribution to employment city preferences.

## Results

### Trends in students' employment city preferences (2016–2020)

Students' employment city preferences exhibit a relatively consistent pattern from 2016 to 2020 (Fig 1). Notably, nearly half of the students choose second-tier cities over the five years, which may suggest a broader pattern in early-career decision-making. The proportion of students selecting first-tier cities slightly increases between 2016 and 2018, followed by a decline. While smaller cities remain the least popular option overall, their proportion gradually increases to 12.97% by

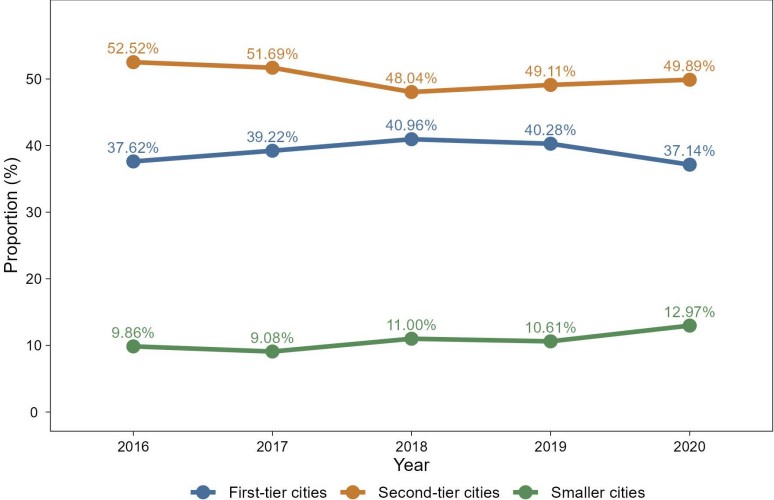

**Fig 1. Changes in employment city preferences of college students in 2016–2020.** Colored lines represent the proportion of students preferring employment in first-tier (blue), second-tier (orange), and smaller (green) cities from 2016 to 2020. Sample sizes for each year are $N = 9040$ (2016), $N = 11888$ (2017), $N = 12477$ (2018), $N = 11544$ (2019), and $N = 5318$ (2020).

2020. This shift may indicate adaptive behavioral adjustments in response to changing labor market conditions or external disruptions such as the COVID-19 pandemic [37].

Employment preferences also vary distinctly by university type (Fig 2). Students from Double First-Class universities (particularly Project "985" and "211" institutions) show a strong preference for first-tier cities, with only 4.23% opting for smaller cities. In contrast, students from regular undergraduate colleges prefer second-tier cities (59.05%), followed by smaller (17.43%) and first-tier (23.52%) cities. Higher vocational institution students display a similar pattern, with the highest proportion in second-tier cities (53.52%), followed by 28.39% in first-tier cities, and 18.09% in smaller cities. The concentration of elite students in first-tier cities aligns with the spatial sorting hypothesis at the institutional level, suggesting that institutional prestige functions as a primary sorting signal for accessing to top-tier urban labor markets.

Furthermore, non-elite students exhibit greater flexibility in their employment city preferences. Among students from regular undergraduate colleges, the proportion preferring first-tier cities peak at 33.09% in 2018 before dropping sharply, accompanied by a corresponding rise in preferences for second-tier cities (Fig 3). This turning point around 2018 coincides with aggressive talent recruitment policies in second-tier cities. Such policy interventions may have enhanced the attractiveness of these cities to non-elite students, lending support to a policy-induced preference mechanism by reshaping perceived opportunities. Students from higher vocational institutions, by comparison, show relatively stable preferences between 2016 and 2019, predominantly choosing second-tier cities. However, the preference for smaller cities increases markedly in 2020, while preferences for first-tier cities decline, likely reflecting the impact of the COVID-19 pandemic on labor demand and job mobility.

### Influencing factors

We analyzed three domains of influence: academic performance, family background, and university characteristics, which represent the cognitive, social, and contextual drivers of students' employment location preferences. The multinomial logistic regression model yields a *McFadden's R²* of 0.10 (Table 2), suggesting that these factors jointly explain a meaningful share of the variation in students' city preferences. The complete coefficient estimates for all variables, including the control variables (year, gender, degree level, and geographic origin), are reported in S10 Table.

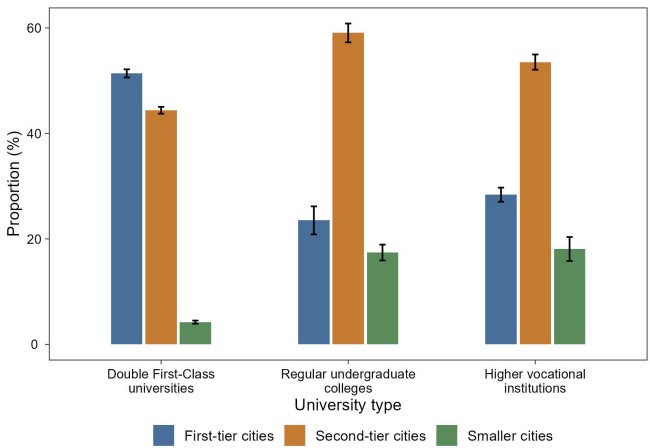

**Fig 2. Employment city preferences by university type.** Colored bars present the mean proportion of students preferring employment in first-tier (blue), second-tier (orange), and smaller (green) cities across three university types. Error bars represent standard errors. Sample sizes are $N=24596$ for Double First-Class universities, $N=13197$ for regular undergraduate colleges, and $N=12474$ for higher vocational institutions.

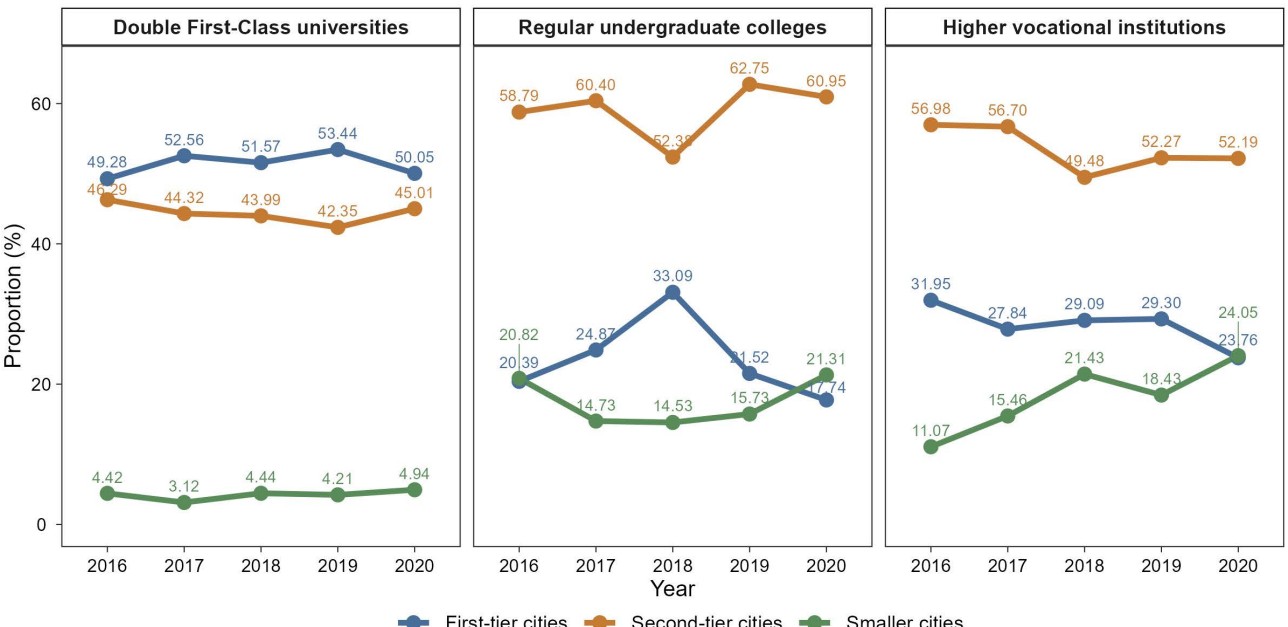

**Fig 3. Changes in employment city preferences by university type.** Colored lines represent the proportion of students from Double First-Class universities (left panel), regular undergraduate colleges (middle panel), and higher vocational institutions (right panel) preferring employment in first-tier (blue), second-tier (orange), and smaller (green) cities.

**Campus performance.** Students' campus performance, including academic performance, leadership experience, and extracurricular participation, is significantly associated with their employment city preferences (Table 2). Notably, higher academic performance correlate with a preference for larger cities. Compared with students who rate their academic performance as "very poor", those reporting "excellent" performance are 2.72 times (odds ratio = 2.72) more likely to prefer first-tier cities and 2.01 times (odds ratio = 2.01) more likely to prefer second-tier cities, relative to smaller cities. Leadership experience also increases the odds of preferring both first and second-tier cities, while participation in extracurricular activities shows a smaller but still positive effect. In contrast, party membership does not significantly influence city preferences once other factors are controlled for. These patterns indicate the spatial sorting mechanism, suggesting that even within the same institutional tier, individual performance serve as an additional productivity signal driving students toward more competitive urban centers.

**Family background.** Family background also plays a distinct role in shaping city mobility preferences (Table 2). Students with an urban *Hukou* have slightly higher odds of preferring first-tier cities (odds ratio = 1.08). Father's education displays a pronounced gradient effect: students whose fathers hold a bachelor's degree or higher are 2.41–3.63 times more likely to prefer first-tier cities and 2.25–2.27 times more likely to prefer in second-tier cities, compared with those whose fathers have only primary education. Household income positively influences city preferences (first-tier: odds ratio = 1.15, second-tier: odds ratio = 1.08). Father's employment in public institutions is not significant, whereas only-child status modestly increases the odds of choosing first- (odds ratio = 1.09) and second-tier cities (odds ratio = 1.13). These findings point toward a compensatory resource mechanism, where family-based capital buffers the costs and risks associated with migrating to larger and more competitive cities.

**University characteristics.** University characteristics exhibit a strong influence on students' employment city preferences (Table 2). Students from Project "985" institutions exhibit the highest odds of preferring first-tier cities, with Project "211" students having 2.56 times higher odds of preferring first-tier over smaller cities. In

**Table 2. Multinomial logistic regression model for employment city preference.**

| Variables | First-tier vs. smaller | Second-tier vs. smaller |
|---|---|---|
| **Campus performance** | | |
| Academic performance (ref. = Very poor) | | |
| Poor | 0.17 (ns) | 0.19 ($p<0.1$) |
| Average | 0.47*** | 0.44*** |
| Good | 0.78*** | 0.64*** |
| Excellent | 1.00*** | 0.70*** |
| Leadership experience (ref. = No) | 0.31*** | 0.18*** |
| Extracurricular participation (ref. = No) | 0.13** | 0.22*** |
| Party membership (ref. = No) | −0.10 ($p=0.10$) | −0.00 (ns) |
| **Family background** | | |
| Urban *Hukou* (ref. = No) | 0.08* | 0.04 (ns) |
| Father's education level (ref. = Primary) | | |
| Junior high school | 0.02 (ns) | 0.39*** |
| High school | 0.19* | 0.42*** |
| Junior college | 0.16 ($p<0.1$) | 0.40*** |
| Bachelor | 0.88*** | 0.82*** |
| Master+ | 1.29*** | 0.81*** |
| Father in public institutions (ref. = No) | −0.04 (ns) | 0.03 (ns) |
| Log annual household income | 0.14*** | 0.08*** |
| Only-child status (ref. = No) | 0.09* | 0.12*** |
| **University characteristics** | | |
| University type (ref. = Project "985" institutions) | | |
| Project "211" institutions | 0.94*** | −0.39*** |
| Regular undergraduate colleges | −1.24*** | −1.07*** |
| Higher vocational institutions | −1.13*** | −1.16*** |

**Notes**: Different values represent standardized coefficients. Sample size: *N* = 50267. Model fit: *Log-Likelihood* = −43132, *McFadden R²* = 0.10, *Likelihood ratio test (χ²)* = 9169.80***. Significance levels: *** $p<0.001$, ** $p<0.01$, * $p<0.05$.

contrast, students from regular undergraduate colleges and higher vocational institutions are significantly less likely than Project "985" students to prefer first-tier (odds ratio = 0.29 and 0.32, respectively) or second-tier cities (odds ratio = 0.34 and 0.31). These results underscore how deeply institutional prestige is tied to city preferences, although the underlying mechanisms, such as prestige signal itself versus the university's geographic location, require further investigation.

 **Relative importance of influencing factors.** To compare the relative importance of these domains, we decompose the grouped effects of campus performance, family background, university characteristics, and control variables. The patterns are consistent across both contrasts (Table 3). University characteristics exhibit the largest average effects, accounting for about 48% of the total absolute effects in both the first-tier vs. smaller and the second-tier vs. smaller comparisons. This highlights the prominent role of institutional-context within the model. Campus performance and family background show more moderate but broadly comparable shares (approximately 14–19%). Control variables, including year, gender, degree level, and geographic origin, also contribute 14–20%. This dominance of university characteristics suggests that institutional signals play a more decisive role than either individual performance or family background in shaping students' city preferences.

**Table 3. Grouped effects of factors influencing students' employment city preferences.**

| Group variables | First-tier vs smaller | | Second-tier vs smaller | |
|---|---|---|---|---|
| | Effect | Relative importance (%) | Effect | Relative importance (%) |
| Campus performance | 0.42 | 18.18 | 0.34 | 18.68 |
| Family background | 0.32 | 13.78 | 0.35 | 19.10 |
| University characteristics | 1.10 | 47.58 | 0.87 | 48.31 |
| Controlled Variables | 0.47 | 20.45 | 0.25 | 13.90 |

**Notes**: Effect represents the mean absolute value of all standardized coefficients in each group.

### Inter-annual variation in the relative importance of influencing factors

We track how the relative influence of these domains shifted across city tiers from 2016 to 2020 (Fig 4). While university characteristics consistently remain the strongest predictor, their influence exhibits a rise–fall–rise trend in both first- and second-tier cities, turning notably after 2017. This pattern may be associated with talent recruitment policies in second-tier cities, which temporarily reduce the differentiating power of top universities. Conversely, family background generally displays an opposite trend. This interannual variation suggests a shifting balance between sorting and compensatory mechanisms: when institutional signals weaken under policy intervention, family resources become more influential in shaping students' migration preferences. Campus performance shows smaller fluctuations over time, but its relative importance increases in 2018 for both first- and second-tier cities. This likely reflects a labor market oversupply, which compels employers to rely more on individual performance signals, such as academic achievement or leadership experience, when evaluating students from the same university tier.

## Discussion

### Spatial sorting through institutional signals

Our findings reveal a pattern of spatial sorting in Chinese students' employment city preferences from 2016 to 2020. Second-tier cities remain the most attractive destinations, serving as a stable anchor for early-career mobility. While

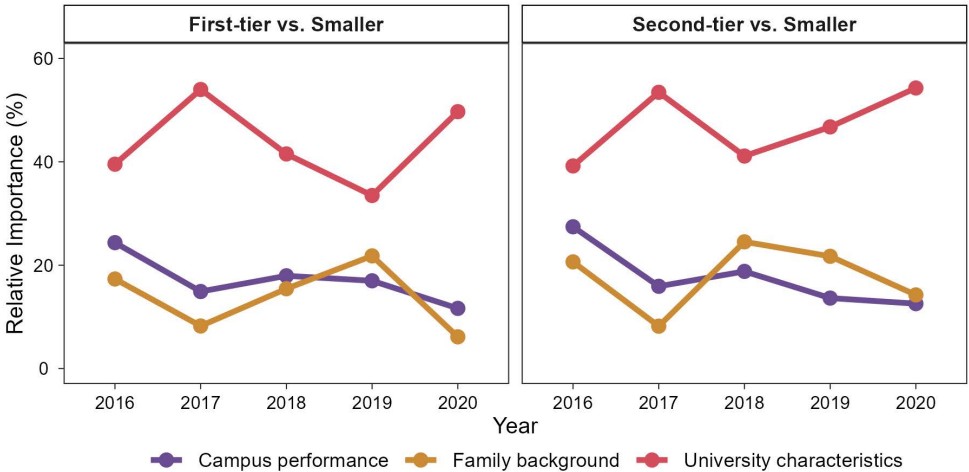

**Fig 4. Interannual variation in the relative importance of factors influencing employment city preferences in 2016–2020.** Colored lines represent the relative importance of campus performance (purple), family background (orange), and university characteristics (red) for first-tier (left panel) and second-tier cities (right panel), relative to smaller cities.

first-tier cities continue to play an important role in the urban hierarchy, their temporary rise followed by a decline may reflect intensifying competition, higher entry barriers, or diminishing marginal advantages as living costs and labor market pressures mount [38–40]. Meanwhile, the rising preferences for smaller cities, especially in 2020, indicate how students adapt to macroeconomic uncertainty and mobility constraints, including those associated with the COVID-19 pandemic [41]. Taken together, these trends suggest that early-career location preferences are shaped by a complex interplay between established urban hierarchies and shifting institutional and economic conditions.

Over the five-year study period, university characteristics consistently exert the strongest and most stable influence on employment city preferences. This finding supports a spatial sorting mechanism driven by institutional prestige and networks, which provide powerful and legible signals of productivity to employers [31,42]. The observed rise–fall–rise pattern in how university characteristics impact preferences for first- and second-tier cities coincides with the periods of intensified local talent recruitment policies. This temporal alignment suggests that such policies may have temporarily weakened the relative advantage of elite institutions, aligning with prior studies which argue that the effects of such talent initiatives are often limited in duration or scope, particularly for students from "First-Class" universities [43,44]. Conversely, students from non-elite institutions appear more responsive to incentives such as housing subsidies and Hukou reforms. Rather than focusing solely on immediate welfare, these students may place greater emphasis on long-term urban integration [45]. In this sense, local policy interventions may temporarily reshape the strength of institutional sorting by altering the structure of perceived opportunities [46].

### Individual performance as a conditional sorting mechanism

Campus performance, including academic achievement and leadership experience, also shapes students' employment city preferences. Stronger individual performance associate with a higher likelihood of preferring first- and second-tier cities, reflecting greater confidence in employability and a higher tolerance for competitive labor markets [47]. However, across the five-year period, the relative influence of individual performance generally declined, with only a slight increase in 2018. This pattern may reflects that institutional prestige often conveys a more powerful signal to employers than individual achievements [32]. Individual performance therefore operates as a conditional or secondary sorting mechanism that differentiates students within similar institutional strata. High-performing students are better positioned to expand their potential city options, particularly in competitive first-tier markets, and to assume greater mobility risks [48,49]. The decreasing relative importance of individual performance may also reflect labor market congestion, where an oversupply of students amplifies institutional signals and narrows the space for individual differentiation. Nevertheless, merit-based indicators remain crucial when students have comparable institutional backgrounds [50], particularly amid increasing transparency in hiring processes and performance-based evaluations [51,52].

### Family background as a compensatory resource mechanism

Family background shows a compensatory role, gaining relative importance when university characteristics are less differentiating. Financial resources, parental education, and household characteristics are positively associated with preferences for larger and more competitive cities. These findings align with a compensatory resource mechanism, whereby family-based economic and social capital helps buffer the costs and risks of migrating to high-cost urban environments [53,54]. Beyond financial support, parents' occupational backgrounds may also shape career aspirations and professional networks [30,55]. Consequently, the economic resources, social capital, and mobility capacity provided by families appear to facilitate students' ability to navigate uncertainty and pursue preferred destinations, especially when institutional prestige alone is insufficient to secure competitive opportunities. Notably, as temporal patterns are inferred from standardized coefficients rather than formally tested through interaction terms, they represent suggestive rather than definitive evidence of a compensatory mechanism.

## Data limitation and policy implications

Although the PSCUS adopts a stratified sampling design and provides broad national coverage of Chinese university students, we must acknowledge several data limitations. First, the publicly available dataset does not include sampling weights or detailed survey design variables (e.g., strata or primary sampling units). As a result, our analyses cannot fully account for the complex survey design. Second, the proportion of Double First-Class (Project "985" and "211") universities in the PSCUS dataset is higher than their actual share within China's higher education system. To assess whether this sampling imbalance affects our findings, we conducted sensitivity analyses. Using national university statistics from the Ministry of Education of the People's Republic of China, we resampled the data to match the real distribution of university types (1% Project "985", 4% Project "211", 39% regular undergraduate institutions, and 56% higher vocational institutions). Across 99 bootstrap iterations, the overall pattern of employment city preferences remained highly consistent (second-tier > first-tier > smaller cities; see S1 Fig). Moreover, multinomial logistic regression models refitted on the resampled datasets also showed that university characteristics retained the largest explanatory power, supporting the robustness of the main conclusions (see S2 Fig).

An additional limitation is that the dependent variable captures students' employment city preferences rather than actual post-graduation locations. Preferences may not fully translate into realized mobility due to labor market constraints, institutional barriers, family considerations, or other unobserved factors. Nevertheless, employment preferences represent a meaningful precursor to mobility, as they reflect how individuals evaluate opportunities and risks prior to labor market entry.

The findings provide several implications for policies aimed at promoting balanced regional development and improving talent allocation. First, the dominant role of university characteristics suggests that strengthening higher education institutions in second-tier and smaller cities, through improved resources, research capacity, and industry collaboration, may enhance their attractiveness to students [56]. Second, although preferences for second-tier cities appear to be growing, continued efforts to improve job matching, career development pathways, and housing support remain important [57,58]. Third, the persistent influence of family background highlights inequalities in access to urban opportunities. Targeted support such as career guidance, internship opportunities, and financial assistance remains critical for students from disadvantaged families [59,60]. Finally, as campus performance continues to shape employment preferences, universities outside first-tier cities should further support students' academic and leadership development help bridge regional disparities in opportunity [61].

Although the observed rise–fall–rise pattern overlaps with the timing of local talent-attraction policies, this alignment does not provide evidence of policy effectiveness. These changes may also reflect broader structural conditions, including fluctuations in labor demand, rising living pressures, and mobility constraints during the COVID-19 pandemic. Given the observational design and the absence of exogenous policy variation, the multinomial logistic models cannot disentangle policy effects from concurrent macro-level developments. Therefore, policy-related interpretations should be understood as suggestive rather than causal. Future research incorporating city-level policy indicators and precise policy timing will be needed to assess these impacts directly.

## Conclusions

This study elucidates the multi-dimensional mechanisms driving Chinese college students' employment city preferences and the evolving structure of early-career talent flows. Second-tier cities consistently attract the largest share of students, while first-tier cities retain dominance among elite students from Double First-Class universities, and smaller cities have gradually gained traction. University characteristics emerge as the most influential factor, underscoring the enduring impact of institutional prestige as a primary spatial sorting mechanism in early career mobility. Academic performance and leadership experience operate as a conditional sorting mechanism within institutional strata. Conversely, family background exerts a compensatory effect by buffering the risks and costs of relocating to competitive urban environments. Our

findings underscore that employment location preferences are shaped by spatial sorting, compensatory resources, and institutional contexts, suggesting that policies aiming to foster balanced urban talent flows should consider institutional, individual, and socio-economic dimensions. Future research should explore how macro-level disruptions, such as the COVID-19, interact with these mechanisms to further influence students' mobility patterns.

## Supporting information

**S1 Fig. Changes in employment city preferences based on bootstrap-resampled data (2016–2020).**
(TIF)

**S2 Fig. Relative importance of three domains of influencing factors based on bootstrap-resampled multinomial logistic regressions (2016–2020).**
(TIF)

**S1 Table. Comparison of variable distributions before and after missing-data deletion.**
(DOCX)

**S2 Table. Classification of cities into three tiers.**
(DOCX)

**S3 Table. Variance inflation factors (VIF) for independent variables in the multinomial logistic regression model (full sample, 2016–2020).**
(DOCX)

**S4 Table. Year-specific adjusted GVIF values for independent variables (2016–2020).**
(DOCX)

**S5 Table. Coefficient estimates of the multinomial logistic regression model for students' employment city preferences (2016).**
(DOCX)

**S6 Table. Coefficient estimates of the multinomial logistic regression model for students' employment city preferences (2017).**
(DOCX)

**S7 Table. Coefficient estimates of the multinomial logistic regression model for students' employment city preferences (2018).**
(DOCX)

**S8 Table. Coefficient estimates of the multinomial logistic regression model for students' employment city preferences (2019).**
(DOCX)

**S9 Table. Coefficient estimates of the multinomial logistic regression model for students' employment city preferences (2020).**
(DOCX)

**S10 Table. The complete set of coefficient estimates for all variables of the multinomial logistic regression model for employment city preference.**
(DOCX)

## Author contributions

**Conceptualization:** Li Wang, Xian Zhang, Yuxiang Li.

**Data curation:** Li Wang, Xian Zhang.

**Formal analysis:** Xian Zhang, Yuxiang Li.

**Investigation:** Li Wang, Xian Zhang.

**Methodology:** Xian Zhang, Yuxiang Li.

**Validation:** Li Wang, Yifei Wang.

**Visualization:** Yifei Wang, Yuxiang Li.

**Writing – original draft:** Li Wang, Xian Zhang, Yuxiang Li.

**Writing – review & editing:** Yuxiang Li.

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
