## [Decision Letter · Decision Letter 0]

23 Oct 2025

Dear Dr. Li,

Thank you for submitting your manuscript to PLOS ONE. After careful consideration, we feel that it has merit but does not fully meet PLOS ONE’s publication criteria as it currently stands. Therefore, we invite you to submit a revised version of the manuscript that addresses the points raised during the review process.

Please carefully address the concerns of the reviewers.

We look forward to receiving your revised manuscript.

Kind regards,

Zhou Yu, PhD

Academic Editor

PLOS ONE

**Journal Requirements:**

2. Please include captions for your Supporting Information files at the end of your manuscript, and update any in-text citations to match accordingly. Please see our Supporting Information guidelines for more information: http://journals.plos.org/plosone/s/supporting-information .

Reviewers' comments:

Reviewer's Responses to Questions

**Comments to the Author**

1. Is the manuscript technically sound, and do the data support the conclusions?

Reviewer #1: No

Reviewer #2: Partly

Reviewer #3: Yes

Reviewer #4: Partly

2. Has the statistical analysis been performed appropriately and rigorously?

Reviewer #1: No

Reviewer #2: No

Reviewer #3: Yes

Reviewer #4: Yes

3. Have the authors made all data underlying the findings in their manuscript fully available?

Reviewer #1: Yes

Reviewer #2: No

Reviewer #3: Yes

Reviewer #4: Yes

4. Is the manuscript presented in an intelligible fashion and written in standard English?

Reviewer #1: Yes

Reviewer #2: No

Reviewer #3: Yes

Reviewer #4: Yes

Reviewer #1: This study utilizes PSCUS (2016-2020) mixed-cross-section data to explore how individual, family, and school factors influence college students' urban preferences for employment. Understanding individual choice behavior from personal, familial, and organizational institutional perspectives holds significant value. Due to concerns regarding the data used and potential methodological shortcomings, the conclusions drawn in this paper are not evaluated at this time.

1.In this dataset, the proportion of students from Double First-Class (985) and 211 universities appears significantly higher than expected. This seems inconsistent with the actual distribution of students across Chinese universities. Could this discrepancy affect related findings, such as employment city preferences?

2. In the analysis of influencing factors, what is the theoretical basis for urban preference in employment choices? Why should academic performance, family background, and school characteristics be considered?

3. The pairwise comparison approach in Table 3's multiple regression seems wrong. We recommend including all three options in the model simultaneously.

4. Is there any basis for using bundled coefficient analysis in Table 4? Is this approach reasonable? Since this result is also derived from pairwise grouping, it appears problematic.

5. Given the pairwise comparison approach in multiple regression, the results in Figure 3 also require correction.

Reviewer #2: ***According to the topic "Where Talent Flows: Trends and Determinants of Chinese Graduates' City Preferences".

It is noted that no words are related to Talent Flows in the abstract and conclusion. Please revise the conclusion to include related content to the topic.

***Please ensure that Talent Flows, The Trends, and Determinants of City Preference are included in the literature review for definitions and related details, such as what the trends are, and which factors of components of determinants of city preference. Operational definitions are necessary.

***Line 177: Determinants of city choice (influencing factors) are academic performance, family background, and university characteristics. It is required for the operation definitions of these key terms.

***Line 83: After excluding cases with missing values, the final analytical sample includes 58136 respondents. Please check if this study included respondents or papers for analysis. If this study included respondents, it is required to include an IRB or ethical approval statement.

***Please add how to select the papers in this study for analysis, such as purposive sampling, and please add selection criteria.

***Please use "researchers" instead of "we".

***Please ensure that the research questions are addressed in the discussion section.

(1) How have Chinese graduates' city preferences evolved over time?

(2) How has the relative influence of academic performance, leadership experience, 68 family background, and institutional environments changed in shaping these decisions?

Reviewer #3: Dear Authors,

Your paper presents a valuable and technically competent analysis of Chinese graduates’ city preferences using nationally representative longitudinal data. The statistical design and execution are generally sound and consistent with the study’s aims. However, to strengthen the methodological transparency and align fully with PLOS ONE’s standards for analytical rigor, I recommend minor revisions to clarify several statistical procedures and assumptions.

Below are my detailed comments, supported by specific references to your manuscript.

1. Model appropriateness and assumptions

You have correctly employed a multinomial logistic regression model to predict graduates’ employment city preferences, which is suitable for your three-category outcome variable (first-tier, second-tier, smaller cities). The model is described clearly on page 8, lines 138–146:

“We use a multinomial logistic regression model [24] to examine the dependent variable graduates' employment city preferences (first-tier, second-tier, or smaller cities).”

However, I recommend that you explicitly state that the Independence of Irrelevant Alternatives (IIA) assumption was tested or at least considered. Because city choices are likely correlated (e.g., first- and second-tier cities may share similar attractions), this assumption is important for model validity. A brief note confirming the assumption holds, or a robustness check using a nested logit model, would strengthen confidence in your estimates.

2. Model diagnostics and fit reporting

While the regression tables (Table 3, pages 12–15) are comprehensive and well formatted, you do not report any model-fit indicators such as pseudo-R², log-likelihood, AIC/BIC, or likelihood-ratio statistics. These would allow readers to assess the explanatory power and comparative adequacy of your models.

I suggest adding a short paragraph (perhaps immediately after Table 3) reporting at least one measure of overall model fit, e.g.:

“Model 1 achieved a McFadden’s pseudo-R² of 0.21, indicating acceptable explanatory power for behavioral data.”

This addition would improve the analytical transparency expected in a PLOS ONE paper.

3. Multicollinearity and robustness checks

The analysis includes several correlated predictors (e.g., academic performance, university type, and university location), which may inflate standard errors. On page 9, lines 144–146, you mention comparing “grouped effects” but do not report multicollinearity diagnostics.

Please confirm whether variance inflation factors (VIFs) were checked, or note that multicollinearity was not problematic. This can be addressed briefly in the Methods section to demonstrate statistical rigor.

4. Bundled coefficient analysis

The bundled coefficients approach is innovative and appropriate for comparing domain-level influences. However, the description on page 15, lines 206–214 would benefit from clarification about how the bundled coefficients were computed and how standard errors were derived:

“This method aggregates the estimated coefficients of related predictors into a single summary measure…”

Please specify whether you used a bootstrapping procedure or analytical aggregation to obtain standard errors for the bundled coefficients. This small addition would make the method more transparent and reproducible for future researchers.

5. Missing data handling

In page 5, lines 81–83, you write:

“After excluding cases with missing values, the final analytical sample includes 58,136 respondents.”

While this approach is reasonable given the large sample size, please clarify whether the excluded cases were missing at random (MAR) or not. If feasible, you could note that results were robust to the exclusion of missing data, or that missingness was minimal. A brief justification will reassure readers that list-wise deletion did not bias your results.

6. Temporal trends and interaction effects

You note on page 17, lines 221–230 that:

“The impact of campus performance… gradually grew stronger, particularly in differentiating choices between first-tier and second-tier cities.”

This is an insightful observation, but it is not statistically tested. You might consider including interaction terms between key predictors (e.g., academic performance × year) or at least acknowledge in the Discussion (around page 18, line 253) that temporal trends were inferred descriptively rather than through formal interaction testing. This clarification would make your interpretation more rigorous without requiring additional analyses.

7. Presentation and interpretation

The interpretation of coefficients is largely accurate, and the reporting of standard errors and significance levels (Table 3) is clear. I appreciate that you avoid over-claiming causality. However, at a few points—such as page 19, lines 268–271, where you write:

“These patterns illustrate a layered decision-making process where graduates weigh symbolic validation, pragmatic incentives, and personal resources under conditions of uncertainty.”

consider softening causal phrasing to indicate association rather than causation, e.g., “the results suggest that graduates may weigh…”.

This will align your interpretation with the correlational nature of the model.

Overall, the statistical analysis is appropriate, competently executed, and largely rigorous.

Your models, sample, and analytical framework are technically sound and consistent with behavioral decision-making research. The requested revisions are minor clarifications aimed at improving transparency and meeting PLOS ONE’s high standards for statistical reporting.

Recommended actions:

Confirm or test the IIA assumption (Methods).

Add model-fit statistics (after Table 3).

Clarify bundled coefficient computation and missing-data handling.

Reframe temporal changes as descriptive unless formally tested.

Adjust phrasing to avoid implied causality.

These adjustments will strengthen the paper’s methodological credibility without altering your results or conclusions.

Reviewer #4: The manuscripts engages an important question for the discipline regarding higher-education graduates' migration decisions. It engages with these questions in an important research site (China) with relevant data (PSCUS) which advances existing research both in the detail level of the dependent variable and by being longitudinal, albeit for a fairly short period of 5 annual waves. The main conclusions are 1) the characteristics of educational institutions matter the most, 2) graduates of non-elite universities are responsive to de-centralization policies, 3) personal traits matter, and 4) family backgrounds matter less, but still matters somewhat. Based on these 4 basic findings the authors offer policy implications for decentralization interventions.

However, the policy implications are mostly based on the second finding, which is the least compelling from a methodological perspective. It relies mostly on annual changes between the first three years in the data and the last two years, which coincide with some new policies. Therefore, the authors see these differences as evidence for the efficacy of said reforms. However, this evidence is completely circumstantial, and it is not clear that the temporal trends cannot be better explained by other factors, most notably COVID19.

Furthermore, while the first finding is very strong, it is actually so strong that it casts doubt on everything else in the manuscript. That is, what the authors describe as a choice of city destination tear, which is very much influenced by "institution's characteristics", may actually be better describes in social rather than geographical terms. Rather than choosing cities, graduates may choose status mobility destinations. University location and type are the most important independent variables by a large margin. So much so that (to this reviewer) it makes little sense to continue the analysis without examining the interactions between university location and other variables. Does university type really matter as much as the analysis suggests, or is it the relative abundance of top-tier institutions in tip tier locations? similarly, do family background, or any other independent variables, have similar associations with graduates destination wishes when they study in top tier cities as in smaller cities? To me such questions seem to drive everything in the manuscript, they could be answered easily by the data, but are ignored in the current version.

**Do you want your identity to be public for this peer review?** For information about this choice, including consent withdrawal, please see our Privacy Policy

Reviewer #1: **Yes:** Shunxu Peng

Reviewer #2: No

Reviewer #3: **Yes:** Holly Carter

Reviewer #4: No

---

## [Author Response · Author response to Decision Letter 1]

21 Nov 2025

REVIEWER COMMENTS TO THE AUTHOR

Reviewer #1:

This study utilizes PSCUS (2016-2020) mixed-cross-section data to explore how individual, family, and school factors influence college students' urban preferences for employment. Understanding individual choice behavior from personal, familial, and organizational institutional perspectives holds significant value. Due to concerns regarding the data used and potential methodological shortcomings, the conclusions drawn in this paper are not evaluated at this time.

>> Thank you very much for your overall assessment and constructive feedback. We have carefully addressed the concerns regarding the data and methodology used in this study. We have: (1) provided a detailed description of the PSCUS (2016–2020) dataset, including sample composition, variables, and operational definitions of key constructs (campus performance, family background, and university characteristics). (2) clarified the analytical approach, including the use of multinomial logistic regression and the interpretation of relative importance measures. (3) conducted additional robustness checks to assess potential biases, such as the overrepresentation of Double First-Class (985/211) universities, by performing bootstrap resamples reflecting the real-world distribution of university types. These checks confirmed that the main results, including patterns of employment city preferences and the relative influence of the three domains remain highly consistent. Please see below for our detailed responses, point by point.

1.In this dataset, the proportion of students from Double First-Class (985) and 211 universities appears significantly higher than expected. This seems inconsistent with the actual distribution of students across Chinese universities. Could this discrepancy affect related findings, such as employment city preferences?

>> Thank you very much for your constructive suggestions. We acknowledge that the proportion of students from Double First-Class (985/211) universities in the PSCUS dataset is higher than their actual share in China’s higher education system. To examine whether this overrepresentation could influence the findings, we conducted additional robustness checks. We have added these robustness checks in the Discussion section (Lines 383–394).

“The proportion of Double First-Class (Project “985” and “211”) universities in the PSCUS dataset is higher than their actual share within China’s higher education system. To assess whether this sampling imbalance may influence the findings, additional robustness checks are conducted. Using national university statistics from the Ministry of Education of the People’s Republic of China, we resampled the data to match the real distribution of university types (1% Project “985”, 4% Project “211”, 39% regular undergraduate institutions, and 56% higher vocational institutions). Across 99 bootstrap iterations, the overall pattern of employment city preferences remained highly consistent (second-tier > first-tier > smaller cities; see S1 Fig). Moreover, multinomial logistic regression models refitted on the resampled datasets consistently showed that university characteristics retained the largest explanatory power, confirming that the study’s core conclusions are robust to variations in sample composition (see S2 Fig).”

2. In the analysis of influencing factors, what is the theoretical basis for urban preference in employment choices? Why should academic performance, family background, and school characteristics be considered?

>> Thank you very much for this helpful comment. We have clarified the theoretical basis in the revised Introduction. The study draws on three complementary theoretical perspectives (Lines 61–73):

“Human capital theory argues that individuals choose destinations that maximize expected returns to their skills and abilities, suggesting that campus performance reflects perceived earning potential and self-assessed competitiveness. Social capital theory proposes how family background shapes risk tolerance, mobility capacity, and career aspirations through economic, cultural, and relational resources. Institutional theory emphasizes how university characteristics, including prestige, location, and embedded networks, expand or constrain students’ access to employment information and opportunities. Together, these perspectives offer a multidimensional framework for understanding how students form city-preference decisions.”

3. The pairwise comparison approach in Table 3's multiple regression seems wrong. We recommend including all three options in the model simultaneously.

>> Thank you very much for this important comment. We have updated our analysis to include all three employment city options simultaneously in the multinomial logistic regression model (see details in the Statistical analysis section, Lines 169–184). The revised results, presented in the updated Tables 2 and 3, show that the main conclusions remain unchanged: university characteristics remain a significant determinant of students’ employment city preferences.

4. Is there any basis for using bundled coefficient analysis in Table 4? Is this approach reasonable? Since this result is also derived from pairwise grouping, it appears problematic.

>> Thank you very much for this insightful comment. The bundled coefficient approach was initially used to combine a group of related predictors into a single composite latent variable, thereby summarizing the collective influence. However, we acknowledge that this approach may introduce ambiguity in the present context. To ensure methodological clarity, we have replaced this approach with a more conventional method based on the mean standardized regression coefficients to assess the effect size and relative importance of each group of variables. The revised methodology is described in the Statistical analysis section (Lines 185–190), and the updated results are presented in Table 3 and Fig 4. Our main conclusions remain consistent under this improved approach.

5. Given the pairwise comparison approach in multiple regression, the results in Figure 3 also require correction.

>> Following the revision of the regression approach, we have updated the figure accordingly (now Fig 4) as well as the corresponding results (Lines 303–317).

Reviewer #2:

According to the topic "Where Talent Flows: Trends and Determinants of Chinese Graduates' City Preferences". It is noted that no words are related to Talent Flows in the abstract and conclusion. Please revise the conclusion to include related content to the topic.

>> Thank you very much for this helpful comment. We have incorporated “talent flows” and related concepts into both the Abstract (Lines 17–18, 30–33) and Conclusions (Lines 419–420, 426–429).

The abstract now highlights that “Employment location choice shapes the flow of talent across urban areas and influences both individual career trajectories and regional human capital distribution” and “These findings highlight the interplay of individual merit, social resources, and institutional prestige in shaping talent flows, offering insights for designing behavioral and policy interventions to support a more balanced and inclusive distribution of talent.”

The conclusion explicitly states that “This study provides insights into the mechanisms shaping Chinese college students’ employment city preferences and the shape of talent flows” and “These findings underscore that employment location decisions are shaped by both structural opportunities and individuals behavioral strategies, suggesting that policies aiming to foster balanced urban talent flows should consider institutional, individual, and socio-economic dimensions.”

***Please ensure that Talent Flows, The Trends, and Determinants of City Preference are included in the literature review for definitions and related details, such as what the trends are, and which factors of components of determinants of city preference. Operational definitions are necessary.

>> Thank you very much for this helpful suggestion. We have incorporated a discussion of talent flows and relevant trends in the revised Introduction (Lines 47–49, 54–54, 80–88):

“These movements constitute important talent flows, commonly defined as the spatial redistribution of highly educated individuals across cities, which influence urban competitiveness, labor market vitality, and long-term economic resilience.”

“Recent studies highlight several major trends in talent flows in China: the enduring pull of first-tier cities, the rising competitiveness of second-tier cities, and the growing influence of talent-attraction policies.”

“To clarify the determinants used in this study, we follow their theoretical foundations and provide operational definitions. Campus performance is measured through academic grades, leadership experience, extracurricular participation, and Party membership, capturing students’ perceived competitiveness and productivity. Family background includes hukou type, father’s education, father’s employment sector, household income, and only-child status, reflecting students’ ability to bear economic costs and navigate uncertainty. University characteristics, including university type and location, capture institutional resources that shape students’ exposure to employment opportunities and expected bargaining power in the labor market.”

***Line 177: Determinants of city choice (influencing factors) are academic performance, family background, and university characteristics. It is required for the operation definitions of these key terms.

>> Thank you very much for this important comment. We have added a dedicated paragraph in the revised introduction to provide operational definitions for the key determinants (Lines 80–88):

“To clarify the determinants used in this study, we follow their theoretical foundations and provide operational definitions. Campus performance is measured through academic grades, leadership experience, extracurricular participation, and Party membership, capturing students’ perceived competitiveness and productivity. Family background includes hukou type, father’s education, father’s employment sector, household income, and only-child status, reflecting students’ ability to bear economic costs and navigate uncertainty. University characteristics, including university type and location, capture institutional resources that shape students’ exposure to employment opportunities and expected bargaining power in the labor market.”

***Line 83: After excluding cases with missing values, the final analytical sample includes 58136 respondents. Please check if this study included respondents or papers for analysis. If this study included respondents, it is required to include an IRB or ethical approval statement.

>> Thank you for pointing this out. This study uses questionnaire survey data from the PSCUS, a nationally representative longitudinal survey conducted by the Institute of Sociology at the Chinese Academy of Social Sciences. All PSCUS data are fully anonymized before release, and our study involved no direct contact with participants. To clarify this, we have revised the Data source section (Lines 107–109).

“As PSCUS is an institutional survey in which all data are fully anonymized and provided for secondary research use, no direct contact with participants occurred in this study.”

In addition, to avoid ambiguity, we have also removed the term “respondents” throughout the manuscript and replaced it with more neutral terms such as “participants”, “observations” or “students”, as appropriate.

***Please add how to select the papers in this study for analysis, such as purposive sampling, and please add selection criteria.

>> Thank you for your suggestion. We would like to clarify that this study analyzes questionnaire survey data rather than papers. The data come from PSCUS, a nationally representative longitudinal survey that uses a stratified sampling strategy administered by the Institute of Sociology at the Chinese Academy of Social Sciences. We did not conduct any additional sampling. This point have now been explicitly stated in the revised Data source section (Lines 101–114).

“The researchers use five waves of data (2016–2020) from the PSCUS (http://www.pscus.cn/), a nationally representative longitudinal survey conducted by the Institute of Sociology at the Chinese Academy of Social Sciences. PSCUS adopts a stratified sampling strategy based on multiple dimensions, including university type (e.g., Double First-Class universities, regular undergraduate colleges, and higher vocational institutions), academic orientation (science and engineering, comprehensive, and humanities and social sciences), and geographic regions. As PSCUS is an institutional survey in which all data are fully anonymized and provided for secondary research use, no direct contact with participants occurred in this study. After excluding cases with missing values, the final analytical sample contains 50267 observations. To evaluate potential bias introduced by missing data, this study compared the distributions of key variables between the raw dataset and the complete-case sample. The differences in proportions are small across variables (less than 4%, S1 Table), indicating no strong systematic patterns of missingness.”

***Please use "researchers" instead of "we".

>> Thank you for your suggestion. In accordance with your recommendation, all instances of “we” referring to the authors have been replaced with “the researchers” or “the authors” in the revised manuscript.

***Please ensure that the research questions are addressed in the discussion section.

(1) How have Chinese graduates' city preferences evolved over time?

(2) How has the relative influence of academic performance, leadership experience, 68 family background, and institutional environments changed in shaping these decisions?

>> Thank you for this helpful comment. We have revised the Discussion section to explicitly address both research questions. we now discuss how students’ city preferences evolve over time, highlighting the sustained attractiveness of second-tier cities, the rise and decline pattern of first-tier cities, and the gradual increase in employment in smaller cities. And we provide a deeper interpretation of the temporal changes in the relative influence of academic performance, leadership experience, family background, and university characteristics. We analyze how institutional prestige shows a rise–fall–rise pattern, how family background exhibits a compensatory dynamic, and how individual performance becomes more or less salient depending on market conditions and policy environments. These expanded discussions appear on Lines 325–381.

Reviewer #3:

Dear Authors,

Your paper presents a valuable and technically competent analysis of Chinese graduates’ city preferences using nationally representative longitudinal data. The statistical design and execution are generally sound and consistent with the study’s aims. However, to strengthen the methodological transparency and align fully with PLOS ONE’s standards for analytical rigor, I recommend minor revisions to clarify several statistical procedures and assumptions.

>> We sincerely appreciate your positive assessment and constructive feedback. In response to the suggestions, we have made several revisions to enhance methodological transparency and clarify statistical procedures and assumptions in line. Specifically, we have: (1) provided more detail on the multinomial logistic regression model, including model specification, coefficient interpretation, model diagnostics, and the approach used to assess the relative importance of influencing factors. (2) included discussion of multicollinearity, variance inflation factors (VIFs), and confirmation that correlated predictors do not substantially bias the results. (3) changed the bundled coefficient approach and explained relative importance calculation method. (4) provided justification for the handling of missing data and its proportion, and confirmed that these did not materially affect the results (5) added clarifications regarding the descriptive nature of temporal trends and the limitations in causal inference, aligning the interpretation with the correlation

---

## [Decision Letter · Decision Letter 1]

23 Dec 2025

Dear Dr. Li,

Thank you for submitting your manuscript to PLOS ONE. After careful consideration, we feel that it has merit but does not fully meet PLOS ONE’s publication criteria as it currently stands. Therefore, we invite you to submit a revised version of the manuscript that addresses the points raised during the review process.

We look forward to receiving your revised manuscript.

Kind regards,

Zhou Yu, PhD

Academic Editor

PLOS One

Journal Requirements:

Additional Editor Comments:

Please address the reviewer's concerns. Thanks.

Reviewers' comments:

Reviewer's Responses to Questions

**Comments to the Author**

Reviewer #5: (No Response)

2. Is the manuscript technically sound, and do the data support the conclusions?

Reviewer #5: Yes

3. Has the statistical analysis been performed appropriately and rigorously?

Reviewer #5: Yes

4. Have the authors made all data underlying the findings in their manuscript fully available?

Reviewer #5: No

5. Is the manuscript presented in an intelligible fashion and written in standard English?

Reviewer #5: Yes

Reviewer #5: Comments to the Authors

Summary

This manuscript uses five waves (2016–2020) of data from the Panel Study of Chinese University Students (PSCUS) to examine trends and determinants of Chinese graduates’ employment city preferences. By classifying cities into first-tier, second-tier, and smaller cities, and estimating multinomial logit models supplemented by a “bundled coefficient” approach, the authors compare the relative roles of campus performance, family background, and university characteristics. The study finds that university type and location are the strongest predictors of city preferences, while the influence of academic performance and leadership experience has become more pronounced over time, and graduates from non-elite institutions are more responsive to talent policies in second-tier cities.

The topic is timely and relevant, the dataset is large and valuable, and the paper is generally clearly written. With some important clarifications and strengthening of the methods and interpretations, this study could make a useful contribution.

Below are some major and minor comments that I hope will help improve the manuscript.

Major Comments

Comments 1: Nature of the data: panel vs. repeated cross-sections

TThe manuscript describes PSCUS as a "nationally representative longitudinal survey" and uses five waves of data, but it is not clear whether the analysis is based on true panel data (the same individuals followed over time) or on repeated cross-sections (different individuals in each wave). This distinction has direct implications for the modelling strategy, because if individuals appear in multiple waves, the independence assumption of the multinomial logit model is violated and standard errors may be underestimated.

Suggestions:

Please clarify the exact structure of the analytic sample. In particular, you may wish to: (1) state explicitly whether individuals are re-interviewed across waves and, if so, how many unique individuals versus person-wave observations are included in the final N; (2) explain how you handle any within-individual correlation, for example by using panel or mixed-effects methods for multinomial outcomes, or by clustering standard errors at the individual level; (3) if instead the data are effectively repeated cross-sections, clarify this and briefly justify why panel methods are not required.

Comment 2: Use of survey design information and national representativeness

The paper characterises PSCUS as a nationally representative survey based on stratified sampling, yet the analysis does not indicate whether survey weights, stratification, or clustering are accounted for. If such design features exist but are not incorporated, this may weaken the claim of national representativeness and could affect the estimated associations.

Suggestions:

It would be helpful to explain more clearly how the survey design is treated in the analysis. For example: (1) indicate whether PSCUS provides sampling weights and design variables, and whether these were used in the descriptive statistics and regression models; (2) if weights or design variables are available but not used, provide a brief rationale and consider softening statements about national representativeness; (3) if no weights are available, please state this explicitly and acknowledge it as a limitation.

Comment 3: Dependent variable - preferences vs. actual choices

The key outcome variable is based on the survey question "Where would you most like to work after graduation?", which captures employment preferences or intentions rather than realised employment locations. However, in several places the manuscript uses terms such as "employment destinations", "city choices", or "job outcomes", which may give the impression that the analysis concerns actual post-graduation locations rather than intended ones.

Suggestions:

To improve conceptual precision, it would be useful to: (1) consistently describe the outcome as employment city preferences or intended employment locations, and make clear early on that this reflects intentions rather than realised behaviour; (2) in the Discussion and Conclusions, soften causal language regarding "flows of talent" or "job outcomes" and frame the findings more explicitly as patterns of preference under conditions of uncertainty; (3) add a sentence or short paragraph in the limitations section noting that intentions may not fully translate into actual behaviour due to labour market constraints, family expectations, or other factors.

Comment 4: Ethics statement and use of human data

In the submission form, the ethics statement is recorded as "N/A", and IRB approval and informed consent are listed as "Not applicable". At the same time, the study analyses individual-level survey data on university students, albeit in anonymized form and as secondary data from PSCUS. PLOS ONE typically expects a clear ethics statement for studies involving human data, even when formal approval is not required or the data are fully anonymized.

Suggestions:

I would encourage you to provide a more explicit ethics description. For example: (1) clarify whether the original PSCUS data collection received ethical approval from an institutional review board or ethics committee, and how informed consent was obtained from participants; (2) if your own institution determined that secondary analysis of anonymized PSCUS data is exempt from additional IRB review, state this explicitly using wording consistent with PLOS ONE guidance, such as "The analysis was based on fully anonymized secondary data; according to the guidelines of [institution], additional ethical approval was not required."; (3) update the Ethics Statement and Methods sections so that the treatment of ethics and consent is transparent to readers.

Comment 5: Data availability and access to PSCUS

There is a small inconsistency between the submission form and the manuscript regarding data availability. The submission form indicates that all data are within the manuscript and its Supporting Information files, whereas the manuscript states that the data are available from http://www.pscus.cn/.

Since PSCUS is an external dataset that typically requires registration or permission, readers may not be able to access the raw microdata directly.

Suggestions:

It would strengthen transparency to harmonise and specify the data availability information. In particular, you may wish to: (1) ensure that the Data Availability Statement in the manuscript and the text entered in the submission system are consistent; (2) clarify what exactly is contained in the Supporting Information (for example, summary tables, derived indicators, or analysis code) and what must be obtained directly from PSCUS (for example, the raw microdata); (3) indicate whether access to PSCUS is open or subject to application or approval, and, if the raw data cannot be shared publicly, follow the PLOS ONE template for restricted third-party data by naming the data owner and explaining how qualified researchers can request access.

Comment 6: Interpretation of multinomial logit coefficients

In the Results and Discussion, some effects are described using phrases such as "increasing the likelihood by X%". For instance, academic performance is said to increase the likelihood of choosing first-tier or second-tier cities by specific percentages. However, the coefficients reported in Table 3 are log-odds, and strictly speaking exp(beta) represents an odds ratio rather than a percentage change in probability. The term "likelihood" is therefore somewhat ambiguous.

Suggestions:

To avoid possible misunderstanding, you might consider: (1) when discussing regression results, either reporting odds ratios (exp(beta)) and explicitly referring to changes in odds, for example "the odds of preferring first-tier cities are X times higher"; (2) alternatively, computing average marginal effects and then describing changes in predicted probabilities, such as "academic performance is associated with a Y percentage point higher probability of preferring first-tier cities"; (3) avoiding expressions that can be interpreted as simple percentage changes in probability unless they are directly based on predicted probabilities from the model.

Comment 7: Bundled coefficient analysis - method and justification

The attempt to group predictors into domains (campus performance, family background, university characteristics) and to compare their relative contribution is conceptually appealing, but the current description of the "bundled coefficients" approach is quite brief. It is not entirely clear how the group-level effect values reported in Table 4 are calculated, whether raw coefficients or transformed values are used, whether they are weighted, and how their standard errors are obtained. The link to Frangioni (2002) also remains somewhat indirect for readers who are unfamiliar with bundle methods in optimisation.

Suggestions:

It would be helpful to elaborate on this method in the Statistical Analysis section. Possible steps include: (1) formally defining how each group’s effect value is constructed from the underlying regression coefficients, including whether absolute values, squared coefficients, or other transformations are used; (2) explaining why this metric is appropriate for comparing "relative importance" across domains and how the associated standard errors are derived; (3) considering references to methodological work that deals directly with the relative importance of groups of predictors, such as R-squared decomposition, dominance analysis, or Shapley value methods, and positioning your approach relative to these; (4) at a minimum, presenting the bundled coefficient results as an exploratory summary of group contributions rather than a precise measure of explanatory power, and acknowledging this in the discussion of limitations.

Comment 8: Causal language around policy interventions

The manuscript links changes after 2018 to intensified "talent war" policies in second-tier cities, including housing subsidies and hukou reforms. This interpretation is plausible and certainly interesting, but in the current design there are no explicit policy variables at the city level in the models, and the evidence relies on descriptive trends over time rather than a causal identification strategy.

Suggestions:

To keep the conclusions aligned with the analytical design, you might: (1) soften statements that imply a direct causal effect of specific talent policies and instead present them as possible explanations or interpretations consistent with the observed patterns, using phrases such as "may be related to" or "is consistent with the interpretation that"; (2) briefly note that identifying the causal impact of concrete policy measures would require a different research design, for example incorporating city-level policy timing in a difference-in-differences framework, and mention this as a promising direction for future research.

Minor Comments

Comments 9: Terminology consistency

The manuscript alternates between expressions such as "employment city choices", "preferences", "performances", and "flows". Using several different terms for the same concept can be slightly confusing, especially since "performances" is more commonly associated with academic outcomes than with location preferences.

Suggestions:

I suggest standardising the terminology as much as possible. For example, you could: (1) select one primary term, such as "employment city preferences" or "intended employment locations", and use it consistently throughout the text; (2) state clearly in the early part of the paper that these preferences refer to intended post-graduation locations rather than realised job placements.

Comment 10: Table and figure labelling

In Table 4, the phrase "employment city performances" appears, which appears to be a typographical slip and may cause confusion. More generally, some readers may benefit from slightly more detailed captions.

Suggestions:

You may wish to: (1) correct "performances" to "preferences" in Table 4; (2) check that all tables and figures are referenced in the main text in numerical order; (3) ensure that each caption is self-contained, briefly indicating the meaning of key variables and units so that readers can understand the content without returning to the main text.

Comment 11: Description of recoding city tiers

Section 2.2.1 notes that response options differed slightly across survey waves and that you recoded them into three tiers, but the details of this recoding are not fully documented. For replication and substantive interpretation, readers may wish to know more about how cities were classified.

Suggestions:

It would improve transparency to add a brief explanation of the recoding. For instance: (1) provide, either in the main text or in an appendix, a list or description of which specific cities fall into the first-tier, second-tier, and smaller city categories; (2) explain how borderline or changing categories were handled when harmonising response options across waves.

Comment 12: Language and style

The English writing is generally clear and readable, but there are a few minor grammatical issues, occasional repetition, and some long sentences that could be streamlined.

Suggestions:

After substantive revisions are complete, a careful language edit would be beneficial. In particular, you may wish to: (1) correct instances such as "employment city performances" to "employment city preferences"; (2) check for consistent singular-plural agreement in phrases like "university characteristics... were/was"; (3) shorten a few very long sentences in the Discussion to improve clarity and flow.

Comment 13: Limitations section

The paper already discusses some limitations, but a few important aspects could be made more explicit to give readers a fuller sense of the study’s scope.

Suggestions:

You might consider adding short statements noting that: (1) academic performance is self-reported rather than based on administrative grade records; (2) the analysis is based on preferences rather than actual employment locations, which may diverge when students enter the labour market; (3) any constraints related to survey weights, complex sampling design, or missing data handling, as discussed in earlier comments, may affect generalisability and should be kept in mind when interpreting the findings.

Overall, I find this study interesting and potentially valuable. Addressing the points above—especially regarding the nature of the data, survey design, ethics, and the interpretation of the models—would substantially strengthen the manuscript.

**Do you want your identity to be public for this peer review?** For information about this choice, including consent withdrawal, please see our Privacy Policy

Reviewer #5: No

---

## [Author Response · Author response to Decision Letter 2]

22 Jan 2026

REVIEWER COMMENTS TO THE AUTHOR

Reviewer #5:

Summary

This manuscript uses five waves (2016–2020) of data from the Panel Study of Chinese University Students (PSCUS) to examine trends and determinants of Chinese graduates’ employment city preferences. By classifying cities into first-tier, second-tier, and smaller cities, and estimating multinomial logit models supplemented by a “bundled coefficient” approach, the authors compare the relative roles of campus performance, family background, and university characteristics. The study finds that university type and location are the strongest predictors of city preferences, while the influence of academic performance and leadership experience has become more pronounced over time, and graduates from non-elite institutions are more responsive to talent policies in second-tier cities. The topic is timely and relevant, the dataset is large and valuable, and the paper is generally clearly written. With some important clarifications and strengthening of the methods and interpretations, this study could make a useful contribution. Below are some major and minor comments that I hope will help improve the manuscript.

>> We sincerely thank you for this positive and encouraging evaluation of our study. We have carefully revised the manuscript to improve clarity, consistency, and transparency. The data structure and methods are now described in greater detail, terminology has been standardized throughout the text, and the limitations section has been expanded to clarify the scope of the analysis and the interpretation of results. Please see below for our detailed responses, point by point.

Major Comments

Comments 1: Nature of the data: panel vs. repeated cross-sections. The manuscript describes PSCUS as a “nationally representative longitudinal survey” and uses five waves of data, but it is not clear whether the analysis is based on true panel data (the same individuals followed over time) or on repeated cross-sections (different individuals in each wave). This distinction has direct implications for the modelling strategy, because if individuals appear in multiple waves, the independence assumption of the multinomial logit model is violated and standard errors may be underestimated.

Suggestions: Please clarify the exact structure of the analytic sample. In particular, you may wish to: (1) state explicitly whether individuals are re-interviewed across waves and, if so, how many unique individuals versus person-wave observations are included in the final N; (2) explain how you handle any within-individual correlation, for example by using panel or mixed-effects methods for multinomial outcomes, or by clustering standard errors at the individual level; (3) if instead the data are effectively repeated cross-sections, clarify this and briefly justify why panel methods are not required.

>> Thank you very much for this important comment. PSCUS is longitudinal in design, and some students may participate in multiple survey waves. However, the publicly available dataset used in this study is fully anonymized and does not include individual identifiers that allow respondents to be linked across waves. As a result, it is not possible to determine how many individuals are re-interviewed in multiple waves or to model within-individual correlation using panel or mixed-effects multinomial models. Given this data limitation, we treat the observations as pooled person-wave data and focus on population-level associations rather than individual-level trajectories, panel methods are therefore not applicable in this setting. To assess whether the pooled estimates are sensitive to temporal variation, we conducted additional multinomial logistic regressions separately for each survey wave. The direction and relative magnitude of key predictors remain largely stable across years (Tables S5-S9), suggesting that the main findings are not driven by any single wave.

We have clarified the structure of the analytic sample and the rationale for the modeling strategy in the revised Data source and Statistical analysis sections (Lines 101–115, 170–187).

Comment 2: Use of survey design information and national representativeness. The paper characterizes PSCUS as a nationally representative survey based on stratified sampling, yet the analysis does not indicate whether survey weights, stratification, or clustering are accounted for. If such design features exist but are not incorporated, this may weaken the claim of national representativeness and could affect the estimated associations.

Suggestions: It would be helpful to explain more clearly how the survey design is treated in the analysis. For example: (1) indicate whether PSCUS provides sampling weights and design variables, and whether these were used in the descriptive statistics and regression models; (2) if weights or design variables are available but not used, provide a brief rationale and consider softening statements about national representativeness; (3) if no weights are available, please state this explicitly and acknowledge it as a limitation.

>> Thank you very much for this helpful comment. PSCUS employs a stratified sampling design across university type, academic orientation, and geographic regions, and is designed to provide broad national coverage of Chinese university students. However, the publicly available PSCUS dataset does not provide sampling weights or detailed survey design variables (e.g., strata or primary sampling units), which limits the ability to conduct fully design-based, weighted analyses. Accordingly, survey weights are not incorporated in the analyses. To address concerns about sample composition, particularly the overrepresentation of Double First-Class universities, we conducted additional sensitivity analyses by resampling the data to match official national statistics on university types and refitting the models across multiple bootstrap iterations. The main patterns and conclusions remain robust. We have now explicitly acknowledged the lack of survey weights and design variables as a data limitation in the Data limitation and policy implications section (Lines 387–403) and clarified the treatment of survey design in the Data source section (Lines 101–115).

Comment 3: Dependent variable - preferences vs. actual choices. The key outcome variable is based on the survey question "Where would you most like to work after graduation?", which captures employment preferences or intentions rather than realized employment locations. However, in several places the manuscript uses terms such as "employment destinations", "city choices", or "job outcomes", which may give the impression that the analysis concerns actual post-graduation locations rather than intended ones.

Suggestions: To improve conceptual precision, it would be useful to: (1) consistently describe the outcome as employment city preferences or intended employment locations, and make clear early on that this reflects intentions rather than realised behaviour; (2) in the Discussion and Conclusions, soften causal language regarding "flows of talent" or "job outcomes" and frame the findings more explicitly as patterns of preference under conditions of uncertainty; (3) add a sentence or short paragraph in the limitations section noting that intentions may not fully translate into actual behaviour due to labour market constraints, family expectations, or other factors.

>> Thank you very much for this important clarification. Following this comment, we now consistently describe as “employment city preferences”, and we clarify early in the Materials and methods section that the survey question asks where students would most like to work after graduation (Lines 125–127). We have added a brief paragraph to the Data limitation and policy implications section noting that city preferences may not fully translate into actual behavior due to labor market constraints, institutional barriers, family considerations, or other unobserved factors. At the same time, we emphasize that employment preferences represent a meaningful and theoretically important precursor to actual mobility, as they reflect how individuals evaluate opportunities and constraints across cities prior to labor market entry (Lines 404–410).

Comment 4: Ethics statement and use of human data. In the submission form, the ethics statement is recorded as "N/A", and IRB approval and informed consent are listed as "Not applicable". At the same time, the study analyses individual-level survey data on university students, albeit in anonymized form and as secondary data from PSCUS. PLOS ONE typically expects a clear ethics statement for studies involving human data, even when formal approval is not required or the data are fully anonymized.

Suggestions: I would encourage you to provide a more explicit ethics description. For example: (1) clarify whether the original PSCUS data collection received ethical approval from an institutional review board or ethics committee, and how informed consent was obtained from participants; (2) if your own institution determined that secondary analysis of anonymized PSCUS data is exempt from additional IRB review, state this explicitly using wording consistent with PLOS ONE guidance, such as "The analysis was based on fully anonymized secondary data; according to the guidelines of [institution], additional ethical approval was not required."; (3) update the Ethics Statement and Methods sections so that the treatment of ethics and consent is transparent to readers.

>> Thank you very much for this important suggestion. The study is based on secondary analysis of the PSCUS dataset, which is publicly released for academic research use in fully anonymized form. We did not have access to any identifiable personal information, nor did we have any direct contact with survey participants. While detailed information on the original ethical review process of the PSCUS data collection is not publicly available, the survey was conducted by the Institute of Sociology at the Chinese Academy of Social Sciences as a large-scale institutional research project. According to the ethical guidelines, secondary analysis of fully anonymized data does not constitute human subjects research requiring additional IRB review. We have revised the Institutional Review Board Statement (Lines 488–492).

Comment 5: Data availability and access to PSCUS. There is a small inconsistency between the submission form and the manuscript regarding data availability. The submission form indicates that all data are within the manuscript and its Supporting Information files, whereas the manuscript states that the data are available from http://www.pscus.cn/. Since PSCUS is an external dataset that typically requires registration or permission, readers may not be able to access the raw microdata directly.

Suggestions: It would strengthen transparency to harmonise and specify the data availability information. In particular, you may wish to: (1) ensure that the Data Availability Statement in the manuscript and the text entered in the submission system are consistent; (2) clarify what exactly is contained in the Supporting Information (for example, summary tables, derived indicators, or analysis code) and what must be obtained directly from PSCUS (for example, the raw microdata); (3) indicate whether access to PSCUS is open or subject to application or approval, and, if the raw data cannot be shared publicly, follow the PLOS ONE template for restricted third-party data by naming the data owner and explaining how qualified researchers can request access.

>> Thank you very much for this important point. We have revised the Data availability statement in the manuscript (Lines 496–499) to clarify that the study is based on third-party data from the Chinese University Student Survey (PSCUS). The raw data are not publicly available and cannot be redistributed by us, as data ownership and access are determined by the data provider.

Comment 6: Interpretation of multinomial logit coefficients. In the Results and Discussion, some effects are described using phrases such as "increasing the likelihood by X%". For instance, academic performance is said to increase the likelihood of choosing first-tier or second-tier cities by specific percentages. However, the coefficients reported in Table 3 are log-odds, and strictly speaking exp(beta) represents an odds ratio rather than a percentage change in probability. The term "likelihood" is therefore somewhat ambiguous.

Suggestions: To avoid possible misunderstanding, you might consider: (1) when discussing regression results, either reporting odds ratios (exp(beta)) and explicitly referring to changes in odds, for example "the odds of preferring first-tier cities are X times higher"; (2) alternatively, computing average marginal effects and then describing changes in predicted probabilities, such as "academic performance is associated with a Y percentage point higher probability of preferring first-tier cities"; (3) avoiding expressions that can be interpreted as simple percentage changes in probability unless they are directly based on predicted probabilities from the model.

>> Thank you for this helpful comment. We have revised the Results section to avoid ambiguous expressions. We now consistently report and interpret effects using odds ratios (exp(β)) (Lines 244–285).

Comment 7: Bundled coefficient analysis - method and justification. The attempt to group predictors into domains (campus performance, family background, university characteristics) and to compare their relative contribution is conceptually appealing, but the current description of the "bundled coefficients" approach is quite brief. It is not entirely clear how the group-level effect values reported in Table 4 are calculated, whether raw coefficients or transformed values are used, whether they are weighted, and how their standard errors are obtained. The link to Frangioni (2002) also remains somewhat indirect for readers who are unfamiliar with bundle methods in optimisation.

Suggestions: It would be helpful to elaborate on this method in the Statistical Analysis section. Possible steps include: (1) formally defining how each group’s effect value is constructed from the underlying regression coefficients, including whether absolute values, squared coefficients, or other transformations are used; (2) explaining why this metric is appropriate for comparing "relative importance" across domains and how the associated standard errors are derived; (3) considering references to methodological work that deals directly with the relative importance of groups of predictors, such as R-squared decomposition, dominance analysis, or Shapley value methods, and positioning your approach relative to these; (4) at a minimum, presenting the bundled coefficient results as an exploratory summary of group contributions rather than a precise measure of explanatory power, and acknowledging this in the discussion of limitations.

>> Thank you for this comment. In the revised manuscript, we have expanded the Statistical analysis section to explicitly describe how the grouped effect measures are constructed (Lines 188–193). Specifically, all predictors are first standardized, and the absolute standardized coefficients within each conceptual domain (campus performance, family background, and university characteristics) are then averaged to produce a summary indicator of relative effect magnitude. This measure is not weighted and does not involve variance-based decomposition.

Comment 8: Causal language around policy interventions. The manuscript links changes after 2018 to intensified "talent war" policies in second-tier cities, including housing subsidies and hukou reforms. This interpretation is plausible and certainly interesting, but in the current design there are no explicit policy variables at the city level in the models, and the evidence relies on descriptive trends over time rather than a causal identification strategy.

Suggestions: To keep the conclusions aligned with the analytical design, you might: (1) soften statements that imply a direct causal effect of specific talent policies and instead present them as possible explanations or interpretations consistent with the observed pattern

---

## [Editor Report · Decision Letter 2]

28 Jan 2026

Dear Dr. Li,

Thank you for submitting your manuscript to PLOS ONE. After careful consideration, we feel that it has merit but does not fully meet PLOS ONE’s publication criteria as it currently stands. Therefore, we invite you to submit a revised version of the manuscript that addresses the points raised during the review process.

We look forward to receiving your revised manuscript.

Kind regards,

Zhou Yu, PhD

Academic Editor

PLOS One

Journal Requirements:

Additional Editor Comments :

The revisions address many of the reviewers’ technical concerns, particularly with respect to data structure and terminology standardization.

To further strengthen the manuscript, however, the theoretical framework should be more explicitly anchored in established migration literature. Clarifying how your analysis extends existing debates—and where it departs from them—will help demonstrate the paper’s contribution more clearly.

Drawing on the broader migration and human capital literature, the hypotheses could be framed through various lenses. First, a Spatial Sorting Hypothesis, grounded in human capital theory, posits that high-achieving students are disproportionately attracted to first-tier cities to maximize returns on skills, with institutional prestige functioning as a dominant labor-market signal. Second, a Compensatory Resource Hypothesis, derived from social capital theory, suggests that family background—through economic and relational resources—buffers the high risks and costs of migration to competitive urban centers, particularly for graduates without elite institutional credentials. Third, a Policy-Induced Preference Hypothesis, informed by choice architecture literature, proposes that local interventions such as hukou reforms and housing subsidies reshape perceived opportunity structures, making second-tier cities more attractive alternatives for non-elite graduates. While these merely are some of the possibilities, it is crucial to link this research to the bigger migration literature.

The manuscript’s contributions should be articulated more explicitly. First, unlike the predominantly cross-sectional literature, this study leverages five waves of longitudinal data to show how urban preferences evolve in response to macro-level shocks, including the COVID-19 pandemic and shifting labor demand—an important and novel contribution. Second, it provides systematic evidence that the “Double First-Class” university designation produces a stratified migration hierarchy in which institutional signals often outweigh individual performance in access to top-tier urban labor markets. Third, the findings demonstrate that “talent war” policies have uneven effects across graduate groups: non-elite students are significantly more responsive to local incentives, offering a more nuanced account of how policy interventions shape regional talent allocation.

More clearly foregrounding these theoretical linkages and substantive contributions would substantially increase the manuscript’s impact.

---

## [Author Response · Author response to Decision Letter 3]

9 Feb 2026

Additional Editor Comments

The revisions address many of the reviewers’ technical concerns, particularly with respect to data structure and terminology standardization.

To further strengthen the manuscript, however, the theoretical framework should be more explicitly anchored in established migration literature. Clarifying how your analysis extends existing debates—and where it departs from them—will help demonstrate the paper’s contribution more clearly.

Drawing on the broader migration and human capital literature, the hypotheses could be framed through various lenses. First, a Spatial Sorting Hypothesis, grounded in human capital theory, posits that high-achieving students are disproportionately attracted to first-tier cities to maximize returns on skills, with institutional prestige functioning as a dominant labor-market signal. Second, a Compensatory Resource Hypothesis, derived from social capital theory, suggests that family background—through economic and relational resources—buffers the high risks and costs of migration to competitive urban centers, particularly for graduates without elite institutional credentials. Third, a Policy-Induced Preference Hypothesis, informed by choice architecture literature, proposes that local interventions such as hukou reforms and housing subsidies reshape perceived opportunity structures, making second-tier cities more attractive alternatives for non-elite graduates. While these merely are some of the possibilities, it is crucial to link this research to the bigger migration literature.

The manuscript’s contributions should be articulated more explicitly. First, unlike the predominantly cross-sectional literature, this study leverages five waves of longitudinal data to show how urban preferences evolve in response to macro-level shocks, including the COVID-19 pandemic and shifting labor demand—an important and novel contribution. Second, it provides systematic evidence that the “Double First-Class” university designation produces a stratified migration hierarchy in which institutional signals often outweigh individual performance in access to top-tier urban labor markets. Third, the findings demonstrate that “talent war” policies have uneven effects across graduate groups: non-elite students are significantly more responsive to local incentives, offering a more nuanced account of how policy interventions shape regional talent allocation.

More clearly foregrounding these theoretical linkages and substantive contributions would substantially increase the manuscript’s impact.

>> Thank you very much for your encouraging feedback and for highlighting how to further enhance the impact of our work. We have carefully revised the manuscript by more clearly anchoring our theoretical framework in established migration literature. In response, we have mainly revised the Introduction and Discussion sections to address these comments (Lines 62–76, 328–388):

1. Clarify theoretical framework and hypotheses: We now linking our study to these established migration theories using three complementary hypotheses. Each hypothesis is directly linked to our empirical analyses, explaining how university prestige, individual performance, and family background shape students’ employment city preferences. we have restructured the Introduction and Discussion to foreground three primary hypotheses:

(1) Spatial Sorting Hypothesis: We now frame the influence of institutional prestige as a dominant labor-market signal that dictates a stratified migration hierarchy. We further discuss individual performance as a secondary, conditional sorting mechanism within this hierarchy.

(2) Compensatory Resource Hypothesis: We emphasize the role of family background (social capital) as a buffer for graduates, especially those from non-elite institutions, against the costs and risks of competitive urban centers.

(3) Policy-Induced Preference Hypothesis: We incorporate the lens of “choice architecture” to explain how local interventions (Hukou, housing subsidies) reshape perceived opportunities for non-elite graduates.

(4) We leverage our five-wave longitudinal analysis to discuss how shifts in urban preferences (especially among non-elite students) align with the “choice architecture” created by local talent-attraction policies (e.g., Hukou reforms, housing subsidies).

2. Substantive Contributions: We now highlight the contribution of leveraging five waves of longitudinal data (2016–2020) to observe how students’ employment city preferences evolve under macro-level shocks, including COVID-19 and shifts in labor demand. Our findings suggest that the “Double First-Class” designation plays a stronger role than individual performance in shaping access to first-tier urban markets. We also observe that non-elite students appear more responsive to local incentives, whereas elite graduates tend to remain aligned with traditional urban hierarchies. These observations provide insights into the interplay of institutional prestige, individual merit, family resources, and policy interventions in early-career mobility.

3. . Terminology and Theoretical Linkage: We have standardized the terminology throughout the manuscript (e.g., using “spatial sorting”, “compensatory resource”, and “policy-induced preference”). We also revised the Discussion headings to reflect these theoretical linkages, ensuring the analysis linked directly to the broader migration literature.

---

## [Editor Report · Decision Letter 3]

12 Feb 2026

Where talent flows: Trends and determinants of Chinese students’ city preferences

PONE-D-25-52295R3

Dear Dr. Li,

We’re pleased to inform you that your manuscript has been judged scientifically suitable for publication and will be formally accepted for publication once it meets all outstanding technical requirements.

Kind regards,

Zhou Yu, PhD

Academic Editor

PLOS One

Additional Editor Comments (optional):

The writing still needs polishing.
---

## [Editor Report · Acceptance letter]

PONE-D-25-52295R3

PLOS One

Dear Dr. Li,

I'm pleased to inform you that your manuscript has been deemed suitable for publication in PLOS One. Congratulations! Your manuscript is now being handed over to our production team.

Kind regards,

on behalf of

Dr. Zhou Yu

Academic Editor

PLOS One